# Can changing neighborhoods influence mental health? An ecological analysis of gentrification and neighborhood-level serious psychological distress—New York City, 2002–2015

**Karen A. Alroy**[1☺], **Haleigh Cavalier**[1☺¤a]*, **Aldo Crossa**[1], **Shu Meir Wang**[1], **Sze Yan Liu**[1¤b], **Christina Norman**[2], **Michael Sanderson**[1¤c], **L. Hannah Gould**[1], **Sung woo Lim**[1]

**1** Division of Epidemiology, New York City Department of Health and Mental Hygiene, Bureau of Epidemiology Services, Queens, New York, United States of America, **2** Division of Mental Hygiene, New York City Department of Health and Mental Hygiene, Bureau of Mental Health, Queens, New York, United States of America

☺ These authors contributed equally to this work.
¤a Current address: Department of Population Health, New York University Langone Health, New York, New York, United States of America
¤b Current address: Department of Public Health, College of Education and Human Services, Montclair State University, Montclair, New Jersey, United States of America
¤c Current address: Division of Family Health, Utah Department of Health and Human Services, Office of Maternal and Child Health, Salt Lake City, Utah, United States of America
* haleigh.cavalier@nyulangone.org

## Abstract

Neighborhood conditions influence people's health; sustaining healthy neighborhoods is a New York City (NYC) Health Department priority. Gentrification is characterized by rapid development in historically disinvested neighborhoods. The gentrification burden, including increased living expenses, and disrupted social networks, disproportionally impacts certain residents. To ultimately target health promotion interventions, we examined serious psychological distress time trends in gentrifying NYC neighborhoods to describe the association of gentrification and mental health overall and stratified by race and ethnicity. We categorized NYC neighborhoods as hypergentrifying, gentrifying, and not-gentrifying using a modified New York University Furman Center index. Neighborhoods with ≥100% rent growth were hypergentrifying; neighborhoods with greater than median and <100% rent growth were gentrifying; and neighborhoods with less than median rent growth were not-gentrifying. To temporally align neighborhood categorization closely with neighborhood-level measurement of serious psychological distress, data during 2000–2017 were used to classify neighborhood type. We calculated serious psychological distress prevalence among adult populations using data from 10 NYC Community Health Surveys during 2002–2015. Using joinpoint and survey-weighted logistic regression, we analyzed serious psychological distress prevalence time trends during 2002–2015 by gentrification level, stratified by race/ethnicity. Among 42 neighborhoods, 7 were hypergentrifying, 7 were gentrifying, and 28 were not gentrifying. In hypergentrifying neighborhoods, serious

**Data Availability Statement:** The majority of the data used in this manuscript can be found in the Open Science Framework Data Repository at https://osf.io/szv36. The repository contains a document titled: Gentrifcation_Manuscript_Datasets.doc that explains the contents of the files and original data sources for the Census and American Community Survey data used to classify neighborhoods into gentrification categories and Census Bureau population estimates. The Community Health Survey data used to estimate the prevalence of serious psychological distress are available, albeit limited, at https://www1.nyc.gov/site/doh/data/data-sets/community-health-survey-public-use-data.page. The Community Health Survey Public Use Data is available at the city-wide level. In order to access the UHF level data as used in the manuscript, you would need to contact epidatarequest@health.nyc.gov to arrange a Data Use Agreement.

**Funding:** The author(s) received no specific funding for this work.

**Competing interests:** The authors have declared that no competing interests exist.

psychological distress prevalence decreased among White populations (8.1% to 2.3%, $\beta = -0.77$, $P = 0.02$) and was stable among Black (4.6% to 6.9%, $\beta = -0.01$, $P = 0.95$) and Latino populations (11.9% to 10.4%, $\beta = -0.16$, $P = 0.31$). As neighborhoods gentrified, different populations were affected differently. Serious psychological distress decreased among White populations in hypergentrifying neighborhoods, no similar reductions were observed among Black and Latino populations. This analysis highlights potential unequal mental health impacts that can be associated with gentrification-related neighborhood changes. Our findings will be used to target health promotion activities to strengthen community resilience and to ultimately guide urban development policies.

## Introduction

Healthy neighborhoods are a foundation of good health [1]. This is particularly true regarding mental health, where social cohesion can confer strength, health, and resiliency. This social resiliency from neighborhood ties can benefit residents' mental health, even among residents experiencing poverty or other hardship [2]. However, community connection can be disrupted by gentrification [3–5]. Gentrification is the process of rapid neighborhood economic investment to repair and rebuild homes and businesses accompanied by an influx of middle-class or affluent people, who are often young and White [6]. In many United States cities, gentrification compounds decades of historic neighborhood disinvestment from longstanding racist policies and practices that have led to inequitable urban development. These include but are not limited to discriminatory homeownership practices like redlining, a financial lending practice that denied homeownership loans to low-income or minority populations [7], exclusionary zoning laws, and racial segregation of public housing [8]. Thus, gentrification disproportionately affects racial minorities in highly segregated urban cities, exacerbating neighborhood inequality by race and class [9]. Furthermore, gentrification can worsen racial/ethnic disparities for health overall and stress, in particular [5, 10]. In Philadelphia, for example, non-Hispanic Black residents in gentrifying neighborhoods are 75% more likely to report fair or poor self-rated health compared with counterparts in non-gentrifying neighborhoods [10] and in a separate study, researchers found that the growing presence of White residents and loss of Black residents contributed to greater stress for the remaining Black community [5]. Gentrification has been identified as a perceived neighborhood stressor across diverse NYC communities [11], and gentrification-related displacement has been associated with negative mental health outcomes [12]. In this study, we examined gentrification as an area-based exposure in NYC neighborhoods, and how it relates to neighborhood-level serious psychological distress prevalence time trends.

Scholars have described distinct waves of gentrification in NYC, and in other cities, that have occurred in the past century. The first wave, prior to the 1970s, was highly-localized and small-scale; the second wave, in the 1970s–1989, was linked with deindustrialization, a back-to-the-city movement, and was highly correlated with artist communities [13, 14]. Later-stage gentrification (the third through fifth waves) have been driven by money-making ventures and characterized by urban commodification [15]. A recent analysis of NYC real estate indicated that in 2003 most building purchases were made by individuals, but by 2014 corporations represented the predominant property buyer in nearly all neighborhoods [16]. Landlords that own multiple buildings, such as corporations, have a higher tenant eviction rate per unit compared with landlords who own few units [17]. Our study focuses on neighborhoods that

experienced later-stage gentrification occurring during 2000–2017. While there is no uniform way to measure gentrification [18, 19], we modified a New York University (NYU) Furman Center gentrification metric to classify all NYC neighborhoods as one of three levels of gentrification: hypergentrifying, gentrifying, or not gentrifying [20]. We selected the Furman Center definition to build on this Center's substantial foundation of scholarship on gentrification in NYC, and because the definition relied on multiple economic factors. Gentrification definitions that incorporate social criteria, such as educational attainment and single-parent households, might unfairly suggest that the onus of neighborhood change is placed on neighborhood community members.

We chose to examine serious psychological distress as an indicator of gentrification's impact on mental health based on literature evidence and mechanistic plausibility [12, 21–24]. We chose to use serious psychological distress as it captures a variety of mental health disorders in a single metric at a population level [25]. Gentrification can increase property value and lead to increases in and quality improvements of physical infrastructure such as grocery stores, green spaces, roads, restaurants and other service sector businesses. However, gentrification can also disrupt existing community cohesion, break social support systems, and weaken mental health protective factors particularly for original residents who experience the neighborhood changing around them [5, 22]. Proposed pathways through which gentrification could affect mental health include a reduced number of affordable housing units, increased rents and living costs which lead to increased stress from housing and financial insecurity, a decreased ability to pay for health-promoting expenditures such as food and medicine, a loss of institutional resources and services, as well as social and cultural displacement [23, 24, 26]. We examined serious psychological distress stratified by race/ethnicity, as we expected mental health outcomes related to gentrification to vary by race. For example, in 2013, Black male community members from gentrifying Washington, D.C. neighborhoods described feeling excluded from White spaces and that Black residents' needs were rarely considered. They expressed feeling powerless and fearful of eventual displacement [27]. While displacement is one potential harm of gentrification, our analysis focuses on the mental health of people residing in changing neighborhoods.

In this study we aimed to characterize trends in gentrification and neighborhood-level serious psychological distress over time, and how the relationship varied by racial/ethnic populations. Given the relatedness between racism and gentrification reported in previous literature [28], we hypothesized that the composition of original residents versus new residents, and thus who are exposed to gentrification, varied by race/ethnicity. We explored the validity of this assumption in an ancillary analysis using housing mobility data. We hypothesized that gentrification's impacts to mental health would affect neighborhood populations differently by race/ethnicity, with worsening prevalence trends among Black and Latino populations compared with White populations.

## Materials and methods

We conducted an ecological time series analysis of the prevalence of serious psychological distress during 2002–2015 in NYC neighborhoods, stratified by level of gentrification and race/ethnicity.

### Classifying levels of gentrification

The original NYU Furman Center gentrification metric categorizes neighborhoods into three types: gentrifying, not gentrifying, and high baseline income neighborhoods [20]. To focus our analysis on the most pronounced manifestation of gentrification, we revised gentrifying

neighborhoods into two more granular levels (gentrifying and hypergentrifying). Hypergentrifying and gentrifying neighborhoods had low median neighborhood income (<40th percentile for the city) in 2000. Hypergentrifying neighborhoods experienced ≥100% rent increase from 2000–2017, while gentrifying neighborhoods experienced a greater than median but <100% rent increase from 2000–2017 [20]. Neighborhoods not categorized as hypergentifying or gentrifying were either low-income and did not experience a greater than median rent increase, or they had moderate to high median neighborhood income in 2000 (≥40th percentile for the city) and thus were considered not eligible to gentrify. While the Furman Center categorizes these two neighborhood types separately, to focus our analysis solely on neighborhoods exposed to gentrification, we combined these two neighborhood types as not gentrifying during 2000–2017.

For the purposes of this study, neighborhoods were defined as NYC United Hospital Fund (UHF) neighborhoods [29], which consist of 42 adjoining zip code areas intended to approximate NYC community planning districts, with roughly 200,000 individuals residing within each UHF. Detailed neighborhood profiles are available online from the NYC Health Department [30]. To calculate level of gentrification, median income and average gross rent data were taken from the 2000 Census and 2013–2017 American Community Survey (ACS) 5-year estimates [31]. ACS 5-year estimates were used to increased reliability and precision compared to 1-year or 3-year estimates. Data from the census are available at the Zip-Code Tabulation Area (ZCTA) geographic level and were aggregated to UHF by summing the values for each ZCTA within a given UHF.

## NYC Community Health Survey

Mental health data came from the NYC Community Health Survey, an annual, cross-sectional, representative, telephone survey of NYC adults (aged ≥18 years) that has been conducted since 2002 and modeled after the Centers for Disease Control and Prevention's Behavioral Risk Factor Surveillance System. Details about the Community Health Survey methodology has been published elsewhere [32]. The Community Health Survey uses a stratified random sample to produce representative citywide and neighborhood-level estimates. The UHF is the smallest geography for which representative prevalence estimates can be calculated. Serious psychological distress is a composite variable from the Kessler-6 questions, a validated scale to measure psychiatric morbidity in community settings [25, 33]. The questions asked, "during the past 30 days how often did you feel. . . 1) so sad that nothing could cheer you up 2) nervous 3) restless or fidgety 4) hopeless 5) that everything was an effort 6) and worthless". The response options, all, most, some, a little, or none of the time, were coded from 4 to 0 respectively. The scores were summed and any individual's response that was ≥13 was considered indicative of serious psychological distress, as this cutoff has been demonstrated in previous literature to be a reliable indicator of mental illness [25, 34]. Serious psychological distress was analyzed as a dichotomous variable. The timeframe examined in this study was, in part, shaped by the available outcome data on serious psychological distress. The Kessler-6 questions were asked in the Community Health Survey in the following ten years: 2002, 2003, 2005, 2006, 2008, 2009, 2010, 2012, 2013, and 2015. Variables on serious psychological distress were not included in the 2017 Community Health Survey even though 2017 was the last year of ACS data that we used for neighborhood classification. To categorize neighborhood type we used small geographies, and consequently selected ACS 5-year data (2013–2017). The resulting two-year difference on each end between ACS and Community Health Survey data sources, was considered acceptable since the focus of this work was trend analysis and not annual data analysis.

Data analyses were stratified by race/ethnicity. The race/ethnicity variable was determined by compiling responses from two questions: 1) "are you Hispanic or Latino?" and 2) "which one or more of the following would you say is your race?" The responses to these questions were combined such that the category of Latino includes people of Hispanic or Latino origin, regardless of race. Black, White, and Asian and Pacific Islander racial categories exclude Latino individuals. We stratified analyses in hypergentrifying and gentrifying neighborhoods by race/ethnicity to examine possible differential associations of gentrification on Black and Latino populations. We do not focus on Non-Hispanic Asian and Pacific Islander populations and people who identified as other race because annual prevalence estimates by levels of gentrification for these groups had small sample sizes and wide uncertainty intervals.

We described serious psychological distress overall from 2002–2015 using ten data points, each corresponding to a year these data were collected during the time period. We further analyzed 2015 data, the most recent year this mental health indicator was measured, to understand differences in prevalence across demographic groups. We describe stratified crude estimates of serious psychological distress by age category and age-adjusted prevalence estimates of serious psychological distress by race/ethnicity.

## Time trend analyses

We used two approaches to analyze prevalence trends. First, for each year the question was asked, serious psychological distress prevalence estimates for each UHF were aggregated by level of gentrification summing the values of all UHFs within each gentrification level. For the analysis of trends over time we used a statistical software, known as Joinpoint (National Cancer Institute). The software uses joinpoint regression to identify the presence or absence of segmented trends and inflection points, or joinpoints. Joinpoint software fits the simplest joinpoint model that the data allow. Although the software was developed for cancer trend analyses, Joinpoint has been used for a diversity of analyses including studies on traffic regulations [35], and suicide rates [36]. We fit autocorrelated errors regression models of serious psychological distress prevalence stratified by level of gentrification and race/ethnicity [37] For each neighborhood and race/ethnicity we described the trend shape (segmented vs. non-segmented), as well as the number of joinpoints and segments (1 joinpoint with 2 segments; 2 joinpoints with 3 segments; etc.).

The National Center for Health Statistics recommends using Joinpoint software for trend analysis of survey data only to identify the number and location of joinpoints, but that survey analysis software should be used to analyze record level data to estimate the trend's slope and variance [38]. For our second approach, we used R v3.5.2 with the survey package to account for the complex sample design [39, 40]. We combined individual-level survey data for the ten years where the Kessler-6 questions were asked, stratified by level of gentrification and race/ethnicity, and fit survey-weighted logistic regression models with linear and quadratic terms. We used these models to obtain estimates of slopes (β) and corresponding 95% confidence intervals (CI). We generated orthogonal polynomial coefficients to account for unequal intervals of time in the trend analysis. For each neighborhood and race/ethnicity model we reported the linear and quadratic estimates, 95% CIs, and *P*-values.

## Demographics by level of gentrification

To characterize the differences in hypergentrifying and gentrifying neighborhoods and to assess the changing demographics in each gentrification level, we examined the absolute change in percent composition of age and racial/ethnic groups using Census Bureau annual population estimates from 2000 and 2018 [41]. We hypothesized that typical demographic

changes associated with gentrification (e.g., an increase in young and White residents) would be more pronounced in hypergentrifying neighborhoods compared with gentrifying neighborhoods. We calculated UHF-level population estimates for different races/ethnicities and age categories, and these estimates were aggregated by level of gentrification and rounded to the nearest 1000. The U.S. Census Bureau population estimates categorize race/ethnicity into five categories: Hispanic, non-Hispanic White, non-Hispanic Black, non-Hispanic American Indian and Alaska Native, and non-Hispanic Asian and Pacific Islander.

### Housing mobility by level of gentrification

We used housing mobility data from the 2017 NYC Community Health Survey [32] in an ancillary analysis to further characterize neighborhood differences, specifically to evaluate the assumption that most Black and Latino populations living in hypergentrifying and gentrifying neighborhoods were original residents who lived in the neighborhoods as they underwent gentrification. We calculated the prevalence of people who lived in the same house for the past 5 or more years stratified by level of gentrification and race/ethnicity.

For time trend analyses, statistical significance of trends was determined if regression model estimate *P*-values were <0.05. Pairwise T-tests were used to compare different strata using a reference category for the 2015 Community Health Survey data on serious psychological distress, and 2017 Community Health Survey data on housing mobility. Analyses were conducted in R v3.5.2 with the survey package [39, 40], Joinpoint software v4.8.0.1 [37], SAS EG v7.15 and SUDAAN v11.0.1. The NYC Department of Health and Mental Hygiene Institutional Review Board determined this work to be secondary research, and as such, exempt from review (19–018).

## Results

### Neighborhood classification

Of the 42 NYC UHF neighborhoods, 7 were classified as hypergentrifying, 7 as gentrifying, and the remaining 28 as not gentrifying in 2000–2017. Three of the hypergentrifying neighborhoods were in Brooklyn (Greenpoint, Williamsburg-Bushwick, Bedford Stuyvesant-Crown Heights), two in Manhattan (East Harlem, Washington Heights-Inwood), one in Queens (Long-Island City-Astoria), and one in the Bronx (Hunts Point-Mott Haven). Four of the gentrifying neighborhoods were in Brooklyn (Sunset Park, East Flatbush-Flatbush, Borough Park, East New York), and three were in the Bronx (High Bridge-Morrisania, Pelham-Throgs Neck, Crotona-Tremont) (Fig 1). A summary of the gentrification criteria by neighborhood can be found in S1 Table.

### NYC Community Health Survey

In the ten years that questions on serious psychological distress were asked in the Community Health Survey, sample size each year ranged from 7,554–10,172. Citywide serious psychological distress fluctuated from 6.4% (95% CI 5.8–7.0) in 2002 to 4.4% (95% CI 3.8–5.2) in 2010, and most recently in 2015 was 5.4% (95% CI 4.8–6.1). In 2015, serious psychological distress prevalence estimates were similar when stratified by age category: 18–24 years old (3.8%, 95% CI 2.5–5.7, *P* = 0.09), 25–44 (5.5%, 95% CI 4.4–6.7, ref), 45–64 (6.3%, 95% CI 5.3–7.4, *P* = 0.32), and ≥65 (4.9%, 95% CI 3.8–6.3, *P* = 0.50). The age-adjusted prevalence was highest in Latino populations (7.9%, 95% CI 6.7–9.2, *P*<0.01), and similar among Black (4.3%, 95% CI 3.3–5.5, *P* = 0.93) and Asian and Pacific Islander individuals (4.7%, 95% CI 3.0–7.1, *P* = 0.79) when compared with White populations (4.4%, 95% CI 3.4–5.5, ref).

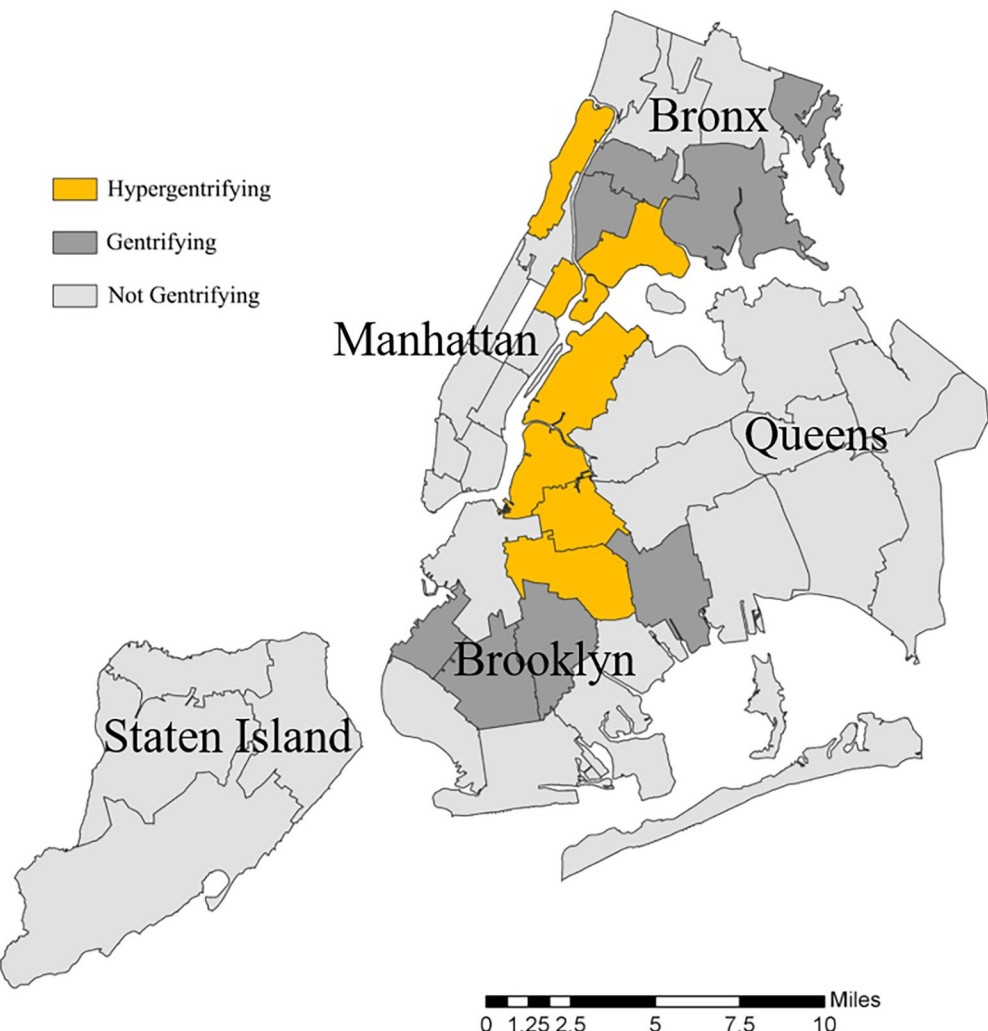

**Fig 1. Map with level of neighborhood gentrification, New York City, 2000–2017.** United hospital fund (UHF) neighborhoods were categorized by level of gentrification using a modified New York University Furman Center definition of gentrification. Data are from the 2000 Census and 2013–2017 American Community Survey (ACS) 5-year estimates.

### Time trend analyses

By level of gentrification, Joinpoint regression models for the prevalence of serious psychological distress in hypergentrifying, gentrifying, and not gentrifying neighborhoods were non-segmented (Table 1). According to linear generalized regression models, the prevalence of serious psychological distress decreased in gentrifying neighborhoods (β = -0.24, $P$ = 0.05), and had minimal change in hypergentrfying (β = -0.13, $P$ = 0.35) and not gentrifying neighborhoods (β = -0.04, $P$ = 0.61). No quadratic regression models fit the data by level of gentrification (Table 1).

By level of gentrification and stratified by race/ethnicity, Joinpoint regression models for the prevalence of serious psychological distress among Black, White, and Latino populations in hypergentrifying, gentrifying, and not gentrifying neighborhoods, were also non-segmented, with one exception in not gentrifying neighborhoods (Table 1, Fig 2). With generalized logistic regression models in hypergentrifying neighborhoods, the prevalence of serious

**Table 1. The joinpoint regression trend shape, and logistic regression model slope, with linear and quadratic terms, of serious psychological distress prevalence by level of gentrification and stratified by race/ethnicity, New York City, 2002–2015.**

| Level of Gentrification | # of Joinpoints | # of Segments | Linear β (95% CI) | *P* value | Quadratic β (95% CI) | *P* value |
|---|---|---|---|---|---|---|
| Hypergentrifying | 0 | 1 | -0.13 (-0.40, 0.14) | 0.35 | 0.26 (-0.01, 0.53) | 0.07 |
| Black | 0 | 1 | -0.01 (-0.5, 0.48) | 0.95 | 0.08 (-0.51, 0.67) | 0.79 |
| Latino | 0 | 1 | -0.16 (-0.47, 0.15) | 0.31 | 0.21 (-0.12, 0.54) | 0.22 |
| White | 0 | 1 | -0.77 (-1.43, -0.10,) | **0.02** | 0.20 (-0.47, 0.87) | 0.55 |
| Gentrifying | 0 | 1 | -0.24 (-0.48, -0.00) | **0.05** | 0.09 (-0.15, 0.33) | 0.46 |
| Black | 0 | 1 | -0.55 (-1.04, 0.06) | **0.03** | -0.05 (-0.58, 0.48) | 0.85 |
| Latino | 0 | 1 | -0.48 (-0.79, -0.17) | **0.00** | 0.11 (-0.22, 0.44) | 0.53 |
| White | 0 | 1 | 0.76 (0.15, 1.37) | **0.01** | 0.35 (-0.16, 0.86) | 0.17 |
| Not Gentrifying | 0 | 1 | -0.04 (0.20, 0.12) | 0.61 | -0.09 (-0.25, 0.07) | 0.27 |
| Black | 1 | 2 | -0.24 (-0.57, 0.09) | 0.16 | -0.55 (-0.98, -0.12) | **0.01** |
| Latino | 0 | 1 | -0.40 (-0.67, -0.13) | **0.00** | 0.16 (-0.13, 0.45) | 0.28 |
| White | 0 | 1 | 0.21 (-0.03, 0.45) | 0.09 | 0.00 (-0.24, 0.24) | 0.99 |

Data from the New York City Community Health Survey

* Non-Hispanic Asian populations and other races and ethnicities are not included due to reliability concerns from small sample sizes.

psychological distress decreased in White populations (β = -0.77, *P* = 0.02), but not in Black (β = -0.01, *P* = 0.95) or Latino populations (β = -0.16, *P* = 0.31) (Table 1). No quadratic regression models fit these data in hypergentrifying neighborhoods. With generalized logistic regression models in gentrifying neighborhoods, the prevalence of serious psychological distress increased in White populations (β = 0.76, *P* = 0.01), and decreased in Black (β = -0.55,

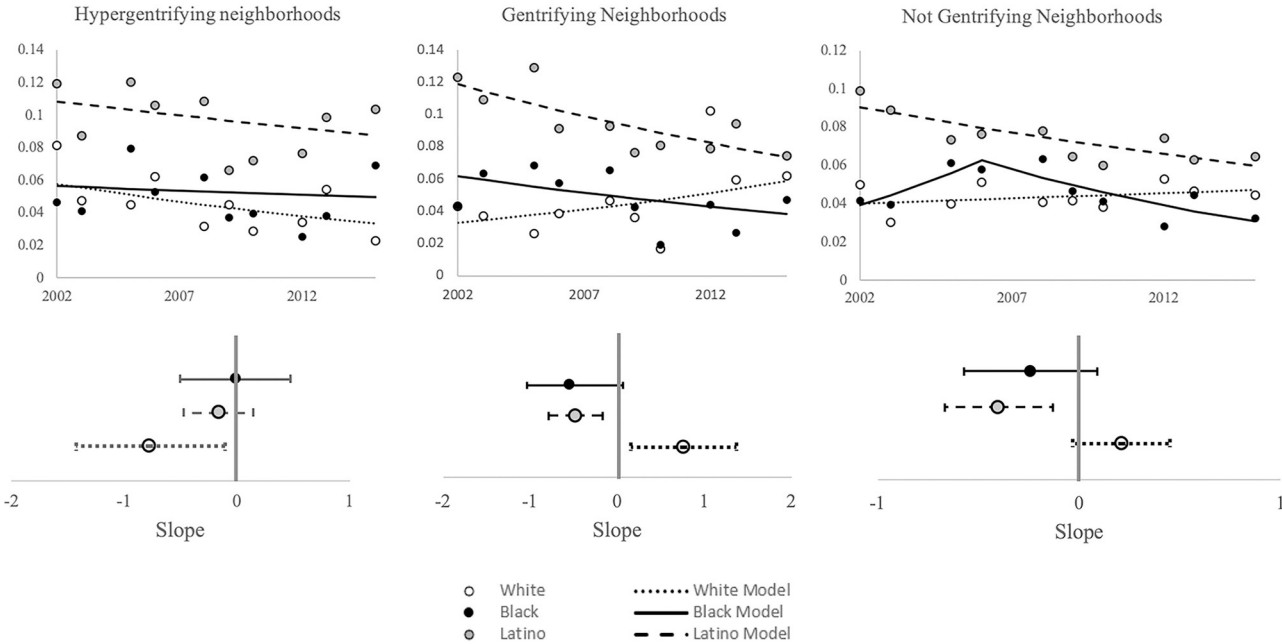

**Fig 2. Time trend analysis of the prevalence of serious psychological distress by level of gentrification and race/ethnicity, New York City, 2002–2015.** These analyses were conducted using Joinpoint autocorrelated errors regression models (Top Row) and generalized logistic regression models (Bottom Row). Data from the New York City Community Health Survey.

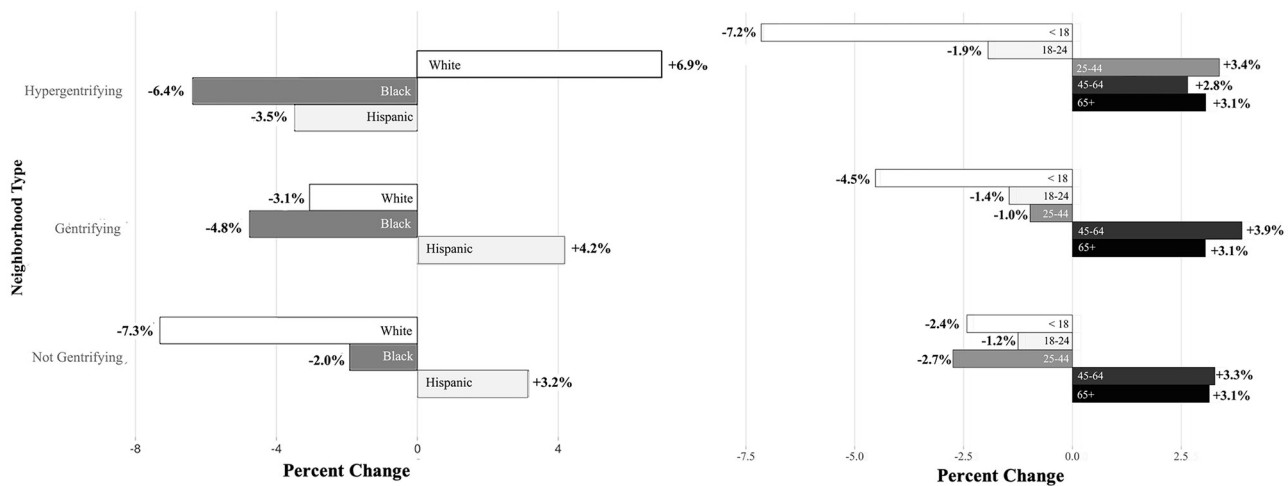

**Fig 3. Hypergentrifying neighborhoods have demographic changes that are different from changes observed in gentrifying and not gentrifying neighborhoods.** Demographic changes in (A) race/ethnicity and (B) age category by level of gentrification, New York City, 2000 and 2018. Data from 2000 and 2018 US Census Bureau interpolated intercensal population estimates.

$P$ = 0.03) and Latino populations (β = -0.48, $P$<0.01) (Table 1). No quadratic regression models fit these data in gentrifying neighborhoods. In not gentrifying neighborhoods, the model for Black populations had one joinpoint and two segments whereas the models for the White and Latino populations were non-segmented. With generalized logistic regression models in not gentrifying neighborhoods, the prevalence of serious psychological distress decreased in Latino populations (β = -0.40, $P$<0.01) and had no change in White (β = 0.21, $P$ = 0.09) and Black populations (β = -0.24, $P$ = 0.16) (Table 1). Corresponding to the segmented regression identified using Joinpoint in not gentrifying neighborhoods the Black population model fit a quadratic regression model ($P$<0.05).

## Demographics by level of gentrification

According to the U.S Census Bureau population estimates from 2000 and 2018 in hypergentrifying neighborhoods, there was an increase in the percentage and total number of people who identified as White (+6.9%, +108,000), and a decrease in the percentage and total number of people who identified as Black (-6.4%, -73,000) and Hispanic (-3.5%, -27,000) (Fig 3). Hypergentrifying neighborhoods also had an increase in the percentage and number of people aged between 25–44 years (+3.4%, +64,000). These same demographic changes were not observed in neighborhoods classified as gentrifying (-3.1% White, -4.8% Black, and +4.2% Hispanic population; and -1.0% people aged 25–44 years) and not gentrifying (-7.3% White, -2.0% Black and +3.2% Hispanic population; and -2.7% people aged 25–44 years) (Fig 3).

## Housing mobility by level of gentrification

In 2017 in hypergentrifying neighborhoods, a higher percentage of Black (70.3%, 95% CI 64.8–75.2, $P$ = 0.002) and Latino (68.5%, 95% CI 63.2–73.4, $P$ = 0.005) people lived in their house for ≥5 years compared with White people (59.6%, 95% CI 53.1–65.8, ref) (Fig 4). White people lived in their house for ≥5 years in similar or greater proportion as compared with Black or Latino people in gentrifying (White: 66.7%, 95% CI 60.7–72.2, ref; Black: 68.1%, 95% CI 63.0–72.8, $P$ = 0.997; Latino: 65.7%, 95% CI 61.4–69.7, $P$ = 0.351) and not

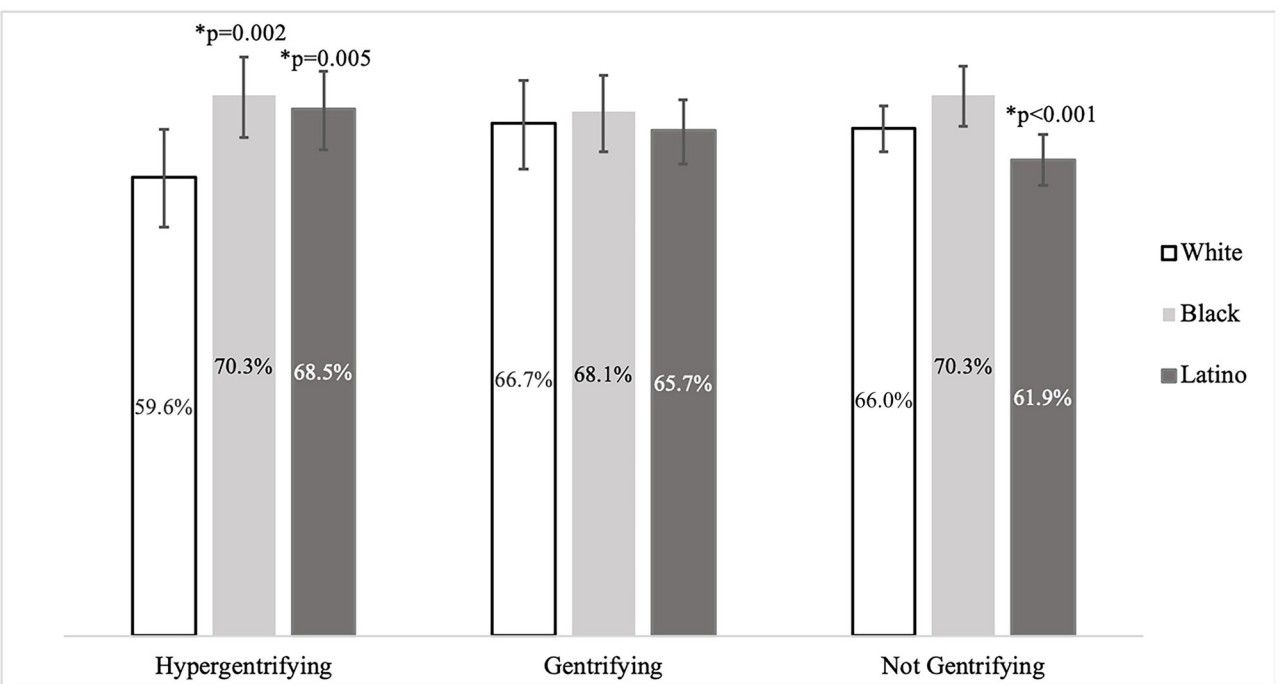

**Fig 4. Hypergentrifying neighborhoods have a higher percentage of Black and Latino people lived in their house for ≥5 years compared with White people.** Percentage of residents who have lived in their homes for ≥5 years stratified by level of gentrification and race/ethnicity in New York City, 2017. Data from the New York City Community Health Survey.

gentrifying neighborhoods (White: 66.0%, 95% CI 63.0–68.9, ref; Black: 70.3%, 95% CI 66.2–74.1, $P = 0.601$; Latino: 61.9%, 95% CI 58.6–65.2, $P<0.001$) (Fig 4).

## Discussion

We examined how gentrification is associated with neighborhood-level mental health over time, and the ways that mental health changes differed among White, Black, and Latino populations living in neighborhoods undergoing gentrification. While our hypothesis that gentrification was associated with worsening mental health prevalence trends among Black and Latino populations compared with White populations was not supported, there was evidence that hypergentrification was associated with increasing health disparities across racial/ethnic groups. In hypergentrifying neighborhoods, White populations had a reduced prevalence of serious psychological distress, with no corresponding reduction for Black and Latino populations. In contrast, in gentrifying neighborhoods, Black and Latino populations experienced reduced serious psychological distress while White populations experienced an increased prevalence trend. In not gentrifying neighborhoods, Latino populations experienced reduced serious psychological distress, while trends were stable for White and Black populations. These findings enhance our understanding of gentrification and health by showing how neighborhood population mental health in NYC has changed over time as neighborhoods themselves have changed, and by reinforcing the notion that gentrification has differential racial/ethnic impacts and can exacerbate health disparities.

We used three-tiers of neighborhood classification to examine gentrification. While both gentrifying and hypergentrifying neighborhoods had low median neighborhood income in the year 2000 and both experienced substantial rent increases, the demographic changes between

the two differed. With Census Bureau population estimate data from hypergentrifying neighborhoods, we found a decreased percentage of Black and Latino residents, and an increased percentage of people aged 25–44 years between 2000 and 2018. Similar demographic patterns were not observed in the gentrifying neighborhoods.

Previous research has found that original residents might benefit from neighborhood gentrification [42], while others suggest that original residents might be harmed by it [21, 43]. The net impacts to health depend on the relative contribution of gentrification benefits versus costs [26]. While gentrification can improve neighborhood infrastructure, enhance safety, and increase economic opportunities, it can also exacerbate income inequality, segregation, discrimination, and displace political power, minimizing the voices and political representation of original residents [28, 44–46]. Gentrification can cause community and cultural displacement and result in heightened policing to "maintain order." All of these changes can lead to increased stress and anxiety among original residents [26]. In participatory research, original neighborhood residents expressed sadness, loss, and feelings of powerlessness in relation to gentrification [11].

We hypothesized that most of the Black and Latino populations in hypergentrifying neighborhoods were original residents and not new residents moving in during the gentrification process. Since Community Health Survey data are annual cross-sectional, and not longitudinal data, it was not possible to identify original residents. Using an approximation, we gained insight into the respondents' residential mobility by examining how long people had lived in their home or apartment. In 2017 in hypergentrifying neighborhoods, there was a higher proportion of Black and Latino residents who had lived in their homes for ≥5 years compared with White residents. We acknowledge that original residents in hypergentrifying neighborhoods were not exclusively Black and Latino, for example, the hypergentrifying Brooklyn neighborhood Greenpoint was historically populated by residents of the Polish community [47]. However, several of the neighborhoods we classified as hypergentrifying are known to have long-standing Black and Latino communities inhabiting them in the late 20[th] and early 21[st] century, including Bushwick [48], Bedford-Stuyvesant [49], Crown Heights [50], and East Harlem [51]. Qualitative data further support our hypothesis. The racial/ethnic changes associated with gentrification are crystalized in an interview from Chiara Valli's 2013 paper when a long-time Brooklyn resident of the Bushwick neighborhood reflected about visibly seeing the demographic and cultural changes that signaled gentrification taking place in his community [48].

Health disparities are deeply entrenched in the United States [1, 52, 53], and this is not the first study to illustrate gentrification exacerbating health disparities by race/ethnicity. The unequal impacts of gentrification on health have been documented among Black populations. An examination of self-rated health in California found no associations overall between gentrification and self-rated health, however, for Black populations, living in a gentrified neighborhood accounted for a 144% increase in the odds of fair or poor self-rated health [43]. Similarly, Huynh et al. found that gentrification was not associated with pre-term birth overall, but that living in a neighborhood undergoing a very high level of gentrification was adversely associated with pre-term birth for people who identified as Non-Hispanic Black [54]. Even the benefits of active green space in gentrifying neighborhoods is not equitably shared among community members, but rather benefits are largely enjoyed by residents who are highly educated and have high incomes [55].

There is limited literature regarding the gentrification exposure on Latino health [56], and additional research is needed. In a review article on gentrification and health, Tulier et al. identify 14 studies acknowledging gentrification's influence on marginalized and underserved populations, of which, most describe impacts to Black residents, and few describe impacts to

Latino and/or Hispanic residents [6]. In NYC, among all racial ethnic groups, Latinos have the highest prevalence of serious psychological distress overall as well as across all levels of gentrification. When stratified by heritage, Community Health Survey data indicate that Puerto Ricans had the highest prevalence of serious psychological distress (12%) among NYC Latino adults, and that Latinos born within the U.S. had a higher prevalence of serious psychological distress compared with Latinos born outside of the U.S. (11% vs 6%) [57]. Trends of serious psychological distress among Latinos decrease in gentrifying and not gentrifying neighborhoods but remain stable in hypergentrifying neighborhoods. These differences in mental health indicators warrant further exploration as community cohesion, acculturation, experiences of racial discrimination, and structural barriers to mental health might vary by neighborhood and level of gentrification. Furthermore, the role of immigration status and racial profiling associated with immigration enforcement on serious psychological distress among Latinos in areas with different levels of gentrification should be examined. While our study does not document mental health harms due to gentrification, we observed that Black and Latino populations in hypergentrifying neighborhoods did not experience the same reduced burden of serious psychological distress as their White counterparts, which may amplify racial mental health disparities in NYC.

We found a segmented regression trend in Black populations in not gentrifying neighborhoods suggesting that there might have been a change in these neighborhoods after 2006 that contributed to improved mental health among Black residents. Not gentrifying neighborhoods was a heterogeneous category, comprised of 25 neighborhoods that had moderate to high baseline income and were not eligible for gentrification and 3 neighborhoods that did not experience greater than median rent increase. Since the focus of this analysis was on gentrification, these not gentrifying neighborhoods were consolidated. Further investigation is warranted to understand the trends of mental health outcomes for not gentrifying neighborhoods, and specifically the underlying cause of the joinpoint that was detected in not gentrifying neighborhoods among Black populations.

The way in which the gentrification exposure is defined has important implications for the associations we observed in this study [18, 19]. We defined two levels of the gentrification exposure based on the magnitude of rent growth during 2000–2017 to identify potential differences in relationships to the outcome across varying degrees of gentrification. We identified differences in demographic composition distinguishing gentrifying and hypergentrifying neighborhoods. Demographic changes observed in hypergentrifying neighborhoods (+6.9% Whites, and -6.4% Blacks) were nearly identical to the demographic changes Sutton et al. described in NYC gentrifying neighborhoods during 2000 and 2010 (+6.3% Whites and -6.8% Blacks) [58]. The neighborhoods we defined as gentrifying were experiencing neighborhood change in increased rent (greater than the citywide median rental growth but <100% rent increase), but they did not experience the typical demographic changes expected with gentrification. Additionally, the prevalence trends of serious psychological distress among hypergentrifying and gentrifying neighborhoods were different overall and when stratified by race/ethnicity. The gentrification exposure definition we used exclusively relied on economic criteria, an approach that avoids any explicit bias that would define a certain neighborhood as containing a younger and potentially healthier population. However, residents in neighborhoods that only experienced rental growth but not demographic changes might not have experienced certain psychosocial aspects of the gentrification exposure, which has been attributed to gentrification driven racial demographic changes [59, 60].

Shifting racial demographics associated with gentrification have been identified as leading to increased preferential treatment of White residents within the neighborhood, causing elevated stress and decreased community cohesion among original residents [11, 56, 59, 61]. Our

findings strengthen evidence that the racial/ethnic demographic changes in addition to substantial rent increases, as seen in our hypergentrifying neighborhoods, are associated with exacerbating health disparities with greater improved health among White populations compared with Black and Latino counterparts. We observed different patterns of serious psychological distress in gentrifying neighborhoods, as compared with hypergentrifying neighborhoods, which experienced less pronounced increase in rent growth and lacked corresponding demographic changes. Specifically, in gentrifying neighborhoods, Black and Latino populations experienced reduced serious psychological distress while White populations experienced an increased prevalence trend. What might be happening in the neighborhoods we classified as gentrifying but not hypergentrifying? Within UHF heterogeneity is one possible explanation. Displacement and gentrification maps created by the Urban Displacement Project suggest that the areas overlapping with the seven UHF neighborhoods we classified as gentrifying are comprised of a variety of different census tracts that are experiencing ongoing gentrification, are at risk of gentrification, are experiencing ongoing displacement of low-income households, and represent some tracts that are not losing low-income households [62]. Additionally, gentrifying and hypergentifying categories may capture gentrification of a different magnitude or stage, which relate to the observed differences in health outcomes. This explanation is supported by the demographic differences observed in hypergentrifying and gentrifying neighborhoods. Community-based participatory research could offer additional qualitative data describing the experiences of residents in these neighborhoods and could help inform interpretation of the mental health outcomes we observed overall and by race/ethnicity in gentrifying neighborhoods.

An ecological study design inherently limits inferences to an aggregated neighborhood-level, and we cannot infer causality of gentrification to individual-level health outcomes in this study. Related, the modifiable areal unit problem, also results from the aggregation of survey data to UHF geography, as the aggregate values are influenced by what geography is chosen [63]. Additionally, the gentrification exposure might not be acting uniformly within neighborhoods, and some neighborhoods might be influenced by gentrification at different times, speeds, and spatial scales. The causally relevant geographic unit important for mental health outcomes is unknown, and mental health may also be influenced by the neighborhoods where people work, attend school, or otherwise spend time aside from their residential neighborhood. Thus, the study is susceptible to the uncertain geographic context problem and spatial polygamy, respectively [64]. We examined gentrification at the UHF level, as this is the smallest geography for which our survey data can provide representative prevalence estimates, however, since gentrification can occur on a hyper-local spatial scale, gentrification might be heterogeneous within a given UHF (Fig 1). Nonetheless, UHFs, and the Community Districts they were designed to approximate, represent meaningful neighborhood boundaries. Residents in these neighborhoods share community board members who advise important decisions on resource allocation, land use, zoning, and other city planning and development projects, all of which are relevant to gentrification [29, 65].

An additional limitation is that cross-sectional data cannot provide evidence of a temporal relationship, and causality cannot be inferred; they simply represent a snapshot of mental health indicators from each year the survey was conducted. Limited sample sizes prohibited us from examining gentrification's impact on the mental health of non-Hispanic Asian populations, other race/ethnicities, and additional subgroupings. We assumed Black and Latino populations represented original neighborhood residents, which does not account for the experiences of new neighborhood residents in gentrifying or hypergentrifying neighborhoods who are Black and Latino. For example, in NYC, Black gentrifiers have been described in the literature in Harlem and Clinton Hill neighborhoods [60]. Time scales and race/ethnicity

classifications of our data sources did not always align, and our demographic analyses represented the differences between two static and arbitrary time points: 2000 and 2018. Nonetheless, neighborhood populations are dynamic with a stream of people constantly moving in and out of neighborhoods.

By measuring gentrification and its associations with mental health, we contextualize current health outcomes with a lens of historic neighborhood disadvantage and systemic racism, which is essential to understanding social determinants of health [52, 53, 66]. While gentrification is an on-going process, few studies have sufficient long-term data to examine health outcomes at multiple time points [6]. The time series study design enables us to examine changing area-level health profiles as neighborhoods undergo change. Some studies examine people who have been displaced by gentrification [67, 68], yet this paper offers an area-based view of neighborhood health and uses race/ethnicity stratification to examine mental health disparities among residents living in changing neighborhoods. Examining serious psychological distress in other jurisdictions experiencing hypergentrification, or at other timepoints in NYC's history, would help elucidate if these findings can be replicated in different times and places.

Exploring the social determinants of mental health, as we did in this analysis, is a priority at the NYC Department of Health and Mental Hygiene [69]. Research and evaluation for policies and programs can be conducted to learn best practices in promoting equitable mental health for all residents in neighborhoods experiencing gentrification [26]. To improve resiliency and reduce mental health disparities in the face of gentrification, policy makers can consider interventions that strengthen safeguards for tenants such as offering legal support for those facing eviction or harassment, and financial assistance for rent; increase access to subsidized housing allowing original residents to remain in their neighborhoods without fear of displacement; and partnerships with community organizations to promote political empowerment for all residents [26]. Future neighborhood developments can engage community residents and offer decision making power to create a vision for neighborhood change that is shared among all residents. The NYC Department of Health and Mental Hygiene, through the Neighborhood Health Action Centers, has mechanisms in place and physical space to implement such community interventions to neighborhoods where gentrification is currently taking place. With locations in East Harlem, Brownsville, and Tremont, and with many established community partners, the Neighborhood Health Action Centers, while not positioned to address the structural causes of inequalities described, can be useful tools in mitigating gentrification-associated health outcomes in the most impacted communities [70]. Further NYC DOHMH gentrification research should include investigating health impacts of policies that directly address structural inequalities.

## Supporting information

**S1 Table. Neighborhood classification, New York City, 2000–2017.**
(DOCX)

## Acknowledgments

Authors thank Charon Gwynn, Gretchen Culp, Pui Ying Chan, Kathleen Reilly, and Katherine Bartley from the New York City Department of Health and Mental Hygiene, Division of Epidemiology.

## Author Contributions

**Conceptualization:** Karen A. Alroy, Haleigh Cavalier, L. Hannah Gould, Sung woo Lim.

**Data curation:** Michael Sanderson.

**Formal analysis:** Haleigh Cavalier.

**Investigation:** Karen A. Alroy, Haleigh Cavalier, Aldo Crossa, Shu Meir Wang, Sze Yan Liu, L. Hannah Gould, Sung woo Lim.

**Methodology:** Karen A. Alroy, Haleigh Cavalier, L. Hannah Gould, Sung woo Lim.

**Project administration:** Karen A. Alroy, Haleigh Cavalier.

**Supervision:** L. Hannah Gould.

**Visualization:** Haleigh Cavalier.

**Writing – original draft:** Karen A. Alroy, Haleigh Cavalier.

**Writing – review & editing:** Karen A. Alroy, Haleigh Cavalier, Aldo Crossa, Shu Meir Wang, Sze Yan Liu, Christina Norman, Michael Sanderson, L. Hannah Gould, Sung woo Lim.

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
