## [Decision Letter · Decision Letter 0]

5 May 2022

PONE-D-21-39108Can changing neighborhoods influence mental health? An ecological analysis of gentrification and neighborhood-level serious psychological distress — New York City, 2002–2015PLOS ONE

Dear Dr. Cavalier,

Thank you for submitting your manuscript to PLOS ONE. After careful consideration, we feel that it has merit but does not fully meet PLOS ONE’s publication criteria as it currently stands. Therefore, we invite you to submit a revised version of the manuscript that addresses the points raised during the review process.

We look forward to receiving your revised manuscript.

Kind regards,

Satya Surbhi, PhD

Academic Editor

PLOS ONE

Journal Requirements:

Reviewers' comments:

Reviewer's Responses to Questions

**Comments to the Author**

1. Is the manuscript technically sound, and do the data support the conclusions?

Reviewer #1: Yes

2. Has the statistical analysis been performed appropriately and rigorously? 

Reviewer #1: I Don't Know

3. Have the authors made all data underlying the findings in their manuscript fully available?

Reviewer #1: Yes

4. Is the manuscript presented in an intelligible fashion and written in standard English?

Reviewer #1: Yes

5. Review Comments to the Author

Reviewer #1: I am not familiar with the exact methods the author's used (joinpoint regression) so I cannot answer whether the methods were appropriate. In general the study addresses an important topic. I think contextualizing the findings and the motivation for the study is an important way the paper can be improved.

6. PLOS authors have the option to publish the peer review history of their article (what does this mean?). If published, this will include your full peer review and any attached files.

Reviewer #1: No

---

## [Author Response · Author response to Decision Letter 0]

29 Aug 2022

Abstract:

The abstract could better set up the argument for the paper by stating up front why there is an expected connection between gentrification and health equity.

• Thank you for this point. We have modified the second and third sentences of the abstract to say “Gentrification is characterized by rapid development in historically disinvested neighborhoods. The burden of gentrification, including increased living expenses, and disrupted social networks, disproportionally impacts certain residents.”

The wording in the sentence starting “During 2000-2017” is a bit awkward. For instance, what is meant by “median and <=100% rent growth? 

• Thank you for pointing this out. We have revised these sentences in the abstract to more clearly describe how neighborhoods were categorized. “We categorized New York City neighborhoods during 2000–2017 as hypergentrifying, gentrifying, and not-gentrifying using a modified New York University Furman Center index. Neighborhoods with >100% rent growth were hypergentrifying; neighborhoods with greater than median and ≤100% rent growth were gentrifying; and neighborhoods with less than median rent growth were not-gentrifying.”

Is joinpoint intended to be capitalized?

• Thank you for catching this. While the software name is Joinpoint (with a capital J), in the abstract here we are referring to Joinpoint regression (with a lowercase j). 

Introduction:

The first two sentences seem a bit disjointed from the rest of the paper. It’s unclear how the neighborhoods experiencing poverty (does this mean neighborhoods with a relatively high percentage of residents with incomes under the poverty line, or neighborhoods where any residents are living in poverty?) is related to the next sentence about gentrification.

• Great point – the text has been updated to add clarity. It now reads: “This is particularly true regarding mental health, where social cohesion can confer strength, health, and resiliency. This social resiliency from neighborhood ties can benefit residents’ mental health, even among residents experiencing poverty or other hardship. However, social cohesion can be disrupted by gentrification (2, 3). Gentrification is the process of rapid neighborhood economic investment to repair and rebuild homes and businesses accompanied by an influx of middle-class or affluent people, who are often young and White (4).”

In the sentence starting “In many United States cities…”, it is odd to state that gentrification “builds upon” neighborhood disinvestment from redlining. Builds upon, has a positive connotation that I don’t think is deserved here. 

• This is a great point. It has been changed from “builds upon” to “compounds”. 

The attribution of gentrification to historic redlining practices is an over simplification. Redlining was only one part of a long history of racist policies and practices that led to uneven urban development and segregation in cities. Therefore, I suggest changing the sentence starting with “due to redlining practices”.

• Another great point. This has been changed and now reads: “In many United States cities, gentrification compounds decades of historic neighborhood disinvestment from longstanding racist policies and practices that have led to inequitable urban development. These include but are not limited to discriminatory homeownership practices like redlining, a financial lending practice that denied home ownership loans to low-income or minority populations (5), exclusionary zoning laws, and racial segregation of public housing (6). Thus, gentrification disproportionately affects racial minorities in highly segregated urban cities, exacerbating neighborhood inequality by race and class (7).”

The focus specifically on psychological distress seems sudden. Perhaps some of the studies reporting on mental health should be cited in the first paragraph rather than the study on self-rated general health.

• Thank you for this suggestion. We have included some description of existing literature regarding gentrification and mental health. We describe that “Gentrification has been identified as a perceived neighborhood stressor across diverse communities in NYC (9), and gentrification-related displacement has been associated with negative mental health outcomes (10).”

What do you mean by “sporadic and small scale” in describing the first wave of gentrification?

• We revised this by describing it as “highly-localized and small-scale”

Why do you choose the 1950’s as the date for the beginning of the first wave? This seems odd since the term gentrification wasn’t termed until the 1960’s.

• We revised this by describing the first wave of gentrification as “prior to the 1970s” as is consistent in the literature.

I think more description of what was happening in the period from 2000-2017 in terms of trends in urban development and economic trends would be useful in making the case for studying this time period specifically. When you say “finance-driven”, do you mean driven by investments from the financial industry, or something else?

• Thank you for these suggestions. We have added additional clarification about what is meant by finance-driven, and we also provided additional context as for why the later stage gentrification time period was the focus of this study. We wrote “Later-stage gentrification (the third through fifth waves) have been driven by money-making ventures and characterized by urban commodification (13). A recent analysis of NYC real estate indicated that in 2003 most building purchases were made by individuals, but by 2014 corporations represented the predominant property buyer in nearly all neighborhoods (14). Landlords that own multiple buildings, such as corporations, have a higher tenant eviction rate per unit compared with landlords who own few units (15).” 

It is not clear why the decision is made to focus specifically on hypergentrifying neighborhoods.

• In our methods section we describe that “to focus our analysis on the most pronounced manifestation of gentrification, we revised gentrifying neighborhoods into two more granular levels” 

• We appreciate the point you’ve raised and added a paragraph in the discussion section, which further explores this categorization. We wrote “In our analysis we used three-tiers of neighborhood classification to examine gentrification. While both gentrifying and hypergentrifying neighborhoods had low median neighborhood income in the year 2000 and both neighborhood types experienced substantial rent increases, the demographic changes in gentrifying and hypergentrifying neighborhoods differed. With Census Bureau population estimate data from hypergentrifying neighborhoods, we found a decreased percentage of Black and Latino residents, and an increased percentage of people aged 25–44 years during 2000–2018. Similar demographic patterns were not observed in the neighborhoods classified as gentrifying.”

In the third paragraph, what is meant by increasing “physical infrastructure”? And what do you include as physical infrastructure? 

• Thank you, this text has been clarified and now reads: “Gentrification can increase property value and lead to increases in and quality improvements of physical infrastructure such as grocery stores, green spaces, roads, restaurants and other service sector businesses.”

It would be nice also to describe the hypothesized pathways that may exist for the relationship between gentrification and mental health distress. In addition to those cited, you might want to review these articles:

Andrew Binet, Gabriela Zayas del Rio, Mariana Arcaya, Gail Roderigues & Vedette Gavin (2021) ‘It feels like money’s just flying out the window’: financial security, stress and health in gentrifying neighborhoods, Cities & Health, DOI: 10.1080/23748834.2021.1885250

Anguelovski I, Cole HVS, O’Neill E, Baro F, Kotsila P, Sekulova F, Pérez del Pulgar C, Shokry G, Garcia Lamarca M, Arguelles Ramos L, Connolly JTT, Honey-Rosés J, Lopez Gay A, Fontán Vela M, Mathaney A, Oscilowicz E, Binet A,Triguero-Mas M. Gentrification pathways and their health impacts on historically marginalized residents in Europe and North America: Global qualitative evidence from 14 cities. Health and Place. 2021 Nov (Vol 72). https://doi.org/10.1016/j.healthplace.2021.102698.

• Great suggestion. While we do not have the space for a full discussion of pathways, we have included the following sentence: “Proposed pathways through which gentrification could affect mental health include increased stress resulting from threats to housing and financial security, loss of institutional resources and services, and social as well as cultural displacement (19, 20).” And we have added the citations that you recommended – these are great papers!

Methods:

Should make clear why you decide to focus on hypergentrifying neighborhoods in the analysis (either here or in the introduction, or across both).

• In the methods we added the clarification that “to focus our analysis on the most pronounced manifestation of gentrification, we revised gentrifying neighborhoods into two more granular levels”

• We have also added a paragraph in the discussion as mentioned above.

The dates included are confusing. It’s clear that the analysis couldn’t use the CHS data from 2017 since the mental health distress scale was not included, but it’s not clear why the census dates did not match the CHS data included (as in ending in 2015), especially since gentrification is the exposure of interest, and not the other way around. This is also confusing since the date 2015 is used in the title, but 2017 is used in other places.

• Thank you for raising this as a confusing point. We tried to provide additional clarification in the methods section. We wrote on page “Variables on serious psychological distress were not included in the 2017 Community Health Survey even though 2017 was the last year of ACS data that we used for neighborhood classification. To categorize neighborhood type we used small geographies, and consequently selected ACS 5-year data (2013-2017). The resulting two-year difference between ACS and Community Health Survey data sources, was considered acceptable since trend analysis and not annual data analysis was the focus of this work.”

Was the decision to use 13 as the cutoff for serious psychological distress based on some precedent or published source?

• Yes, the text has been changed to reflect this and two citations have been added that establish and use this cutoff.

Could you describe a bit more what you mean in the sentence beginning “We analyzed 2015 data, …” What do you mean hear by contextualizing?

• Yes, we have changed the wording for more clarity. The text now reads: “We further analyzed 2015 data, the most recent year this mental health indicator was measured, to understand differences in prevalence across different demographic groups.” 

How was the census data aggregated into UHF areas?

• A sentence was added to address this at the end of the materials and methods section “classifying levels of gentrification.” It reads: “Data from the census are available at the Zip-Code Tabulation Area (ZCTA) geographic level and were aggregated to UHF by summing the values for each ZCTA within a given UHF.”

As someone who does not know anything about joinpoint regression, I think it would be helpful if you could describe what joinpoint regression is and why you chose to use it?

• Thank you for this suggestion. We have expanded on our description of joinpoint regression and Joinpoint software. For the analysis of trends over time we used a statistical software, known as Joinpoint (National Cancer Institute). The software uses joinpoint regression, also known as change point or segmented regression, to identify the presence or absence of segmented trends and inflection points, or joinpoints. Joinpoint software fits the simplest joinpoint model that the data allow. Although the software was developed for cancer trend analyses, Joinpoint has now been utilized for a diversity of analyses including studies on traffic regulations (28), and suicide rates (29). 

Results:

In the section starting with “Demographics by level of gentrification”, and beyond, you refer to census data from 2000-2018 (instead of 2017 as stated earlier). Please clarify the years included.

• The 2017 5-year estimate ACS data were used to classify gentrification levels. The 5-year estimate data were recommended over 1-year or 3-year estimate data since the geography we used was smaller and the 5-year estimate data increase precision. At the time the paper was written, the 2017 data were the most recent of available data sets. The 2018 census population estimate data was used when looking at demographic variables and how they changed over time. We always used the most recent data that were available and although both data sets came from the Census Bureau, they are released at different times. Since the population estimates are released annually, these 2018 data were available at the time of writing. The language in the paper has been changed to reflect more clearly that the two data sets for these analyses are distinct. Data used in the demographics analysis are now always referred to as Bureau population estimates and data used in the gentrification analysis are referred to as 2000 Census/2017 ACS data. We decided to use 2018 data for the demographic analysis as the goal of this ancillary analysis was capture a demographic snapshot at the start and end of the period of gentrification we are focused on in the main analysis and compare the two. 

In the “Demographics by level of gentrification” section, are the relative and absolute changes in demographics averaged over the years included, or do the represent change from 2000 to 2018, not considering the years in between?

• Thank you, upon re-reading this text, we realize “percent change” is confusing and not fully accurate. We used 2000 and 2018 data to look at absolute changes in the percent composition of various age and racial/ethnic groups. For example, if 30% of the population was Black in 2000 and 35% of the population was Black in 2018, the absolute change was +5%. The text in has been updated for clarity and now reads in the Methods Section: “In order to characterize the differences in hypergentrifying and gentrifying neighborhoods and to assess the changing demographics in each level of gentrification, we examined the absolute change in percent composition of age and racial/ethnic groups using Census Bureau annual population estimates from 2000 and 2018 (34).” And in the Results section: “According to the U.S Census Bureau population estimates from 2000 and 2018 in hypergentrifying neighborhoods, there was an increase in the percentage and total number of people who identified as White…”

Discussion:

The reasons why we might see the effects observed in hypergentrified neighborhoods but not gentrified ones should be better indicated and reasoned in the discussion.

• We have added to and emphasized the discussion of this topic in the paper. The text now reads: “What might be happening in the neighborhoods we classified as gentrifying but not hypergentrifying? Within UHF heterogeneity is one possible explanation. Displacement and gentrification maps created by the Urban Displacement Project suggest that the areas overlapping with the seven UHF neighborhoods we classified as gentrifying are comprised of a variety of different census tracts that are experiencing ongoing gentrification, are at risk of gentrification, are experiencing ongoing displacement of low-income households, and represent some tracts that are not losing low-income households (59). Also, gentrifying and hypergentifying categories may capture gentrification of a different magnitude or stage, which relate to the observed differences in health outcomes. This explanation is supported by the demographic differences observed in hypergentrifying and gentrifying neighborhoods. Community-based participatory research could offer additional qualitative data describing the experiences of residents in these neighborhoods and could help inform interpretation of the mental health outcomes we observed overall and by race/ethnicity in gentrifying neighborhoods.”

In a few places, you use direct quotes from participants in other published studies. This seems odd to me.

• This is understandable. We feel as though incorporating quotes from qualitative research studies in the discussion provides relevant context and highlights community opinions and voices on the topic, since we were not able to supplement this work with our own independent qualitative research.

In the paragraph starting “In our analyses…”, it might be useful to describe why you didn’t include length of residency as a criteria for being considered an “original resident”. In addition, you might state more clearly here why you use race/ethnicity as a proxy for being an original resident (which is implied, but could be more directly worded).

• Thank you for this feedback. We have acknowledged why it was difficult to assess who was an “original resident” using the Community Health Survey data available for this analysis. We wrote the following: “In our analyses, we hypothesized that most of the Black and Latino population in hypergentrifying neighborhoods were original residents and not new residents moving in during the gentrification process. Since Community Health Survey data are annual cross-sectional, and not longitudinal data, it was not possible to identify original residents. Using an approximation, we gained insight into the respondents’ residential mobility by examining how long people had lived in their home or apartment. We found that in 2017 in hypergentrifying neighborhoods, there were a higher proportion of Black and Latino residents who had lived in their homes for ≥5 years compared with White residents.” 

In the paragraph starting “health disparities are deeply entrenched in NYC…”, I find it confusing that you bounce between referencing studies from NYC and those from other places (like California). The purpose of the paragraph might be more clear if you refer only to NYC, and also state why NYC might be a unique case. There are other studies which are published about gentrification and health in NYC, for instance this one:

Cole HVS, Triguero-Mas M, Connolly J, Anguelovski I. Determining the Health Benefits of Green Space: Does gentrification matter? Health and Place. 2019; 57, May 2019: 1-11. https://doi.org/10.1016/j.healthplace.2019.02.001

• Thank you for this suggestion. We have refined the topic sentence to say “Health disparities are deeply entrenched in the United States (1, 47, 48), and this is not the first study to illustrate gentrification exacerbating health disparities.” In addition, we have clarified that even the benefits associated with green space in gentrifying neighborhoods is largely unequitable. “Even the benefits of active green space in gentrifying neighborhoods is not equitably shared among community members, but rather benefits are largely enjoyed by residents who are highly educated and have high incomes (51).” 

In the paragraph starting with “We observed different patterns of…”, it is a bit misleading to state that “… we have shown had a less extreme increase in rent growth…” since this is how you defined these neighborhoods and classified them in the first place, rather than a finding from the study.

• Thank you. We have changed the wording. This sentence now reads “We observed different patterns of serious psychological distress in gentrifying neighborhoods, as compared with hypergentrifying neighborhoods, which experienced less extreme increase in rent growth and lacked corresponding demographic changes.”

In the paragraph starting with “An ecological study design inherently…”, you state that 2000 and 2018 are arbitrary dates. It might be more interesting then to also discuss and characterize what type of urban and economic development occurred during this time in NYC. Also, what effect do you think the 2008 recession had on your findings?

• In the introduction we discuss different types of urban and economic development and we describe the patterns of gentrification in NYC. We write “Scholars have described distinct waves of gentrification in NYC, and in other cities, that have occurred in the past century. The first wave, prior to the 1970s, was highly-localized and small-scale; the second wave, in the 1970s–1989, was linked with deindustrialization, a back-to-the-city movement, and was highly correlated with artist communities (11, 12). Later-stage gentrification (the third through fifth waves) have been driven by money-making ventures and characterized by urban commodification (13). A recent analysis of NYC real estate indicated that in 2003 most building purchases were made by individuals, but by 2014 corporations represented the predominant property buyer in nearly all neighborhoods (14). Landlords that own multiple buildings, such as corporations, have a higher tenant eviction rate per unit compared with landlords who own few units (15). This study focuses on neighborhoods that experienced later-stage gentrification occurring during 2000–2017.” Additionally, we did not observe any joinpoints associated with the 2008 recession indicating a change in the existing trend. 

The focus on developing targeted interventions in the last paragraph is a bit misleading and I don’t think links well to the findings from the paper. The paper demonstrates the relationship between structural inequalities and mental health inequalities. But it’s unclear how and if the interventions planned by the health department could then develop interventions that address such structural inequalities, rather than to promote health interventions that merely target individuals or populations that are affected by such inequalities.

• Thank you. This is a great point. The text has been updated to clarify between addressing structural inequalities and mitigating impacts among those most affected. The text now reads: “With locations in East Harlem, Brownsville, and Tremont, and with many established community partners, the Neighborhood Health Action Centers, while not positioned to address the structural causes of inequalities described, can be useful tools in mitigating gentrification-associated health outcomes in the most impacted communities (65). Further NYC DOHMH gentrification research should include investigating health impacts of policies that directly address structural inequalities.”

Acknowledgements:

Check the spelling of Kathleen Reilly’s name

• It was spelled incorrectly and has been fixed, thank you for catching this!

Reviewer #2: I am not familiar with the exact methods the author's used (joinpoint regression) so I cannot answer whether the methods were appropriate. In general the study addresses an important topic. I think contextualizing the findings and the motivation for the study is an important way the paper can be improved.

• Thank you for this feedback. We have helped to contextualize the findings by expanding on the description of the type of gentrification that was occurring during the study period. We wrote “Later-stage gentrification (the third through fifth waves) have been driven by money-making ventures and characterized by urban commodification (13). A recent analysis of NYC real estate indicated that in 2003 most building purchases were made by individuals, but by 2014 corporations represented the predominant property buyer in nearly all neighborhoods (14). Landlords that own multiple buildings, such as corporations, have a higher tenant eviction rate per unit compared with landlords who own few units (15). This study focuses on neighborhoods that experienced later-stage gentrification occurring during 2000–2017. 

• In addition, we also clarified the motivation behind these analyses: to better understand how gentrification might contribute to health inequities and ultimately to improve upon the services provided by the NYC Health Department. We wrote “To improve resiliency and reduce mental health disparities in the face of gentrification, policy makers can consider interventions that strengthen safeguards for tenants such as offering legal support for those facing eviction or harassment, and financial assistance for rent; increase access to subsidized housing allowing original residents to remain in their neighborhoods without fear of displacement; and partnerships with community organizations to promote political empowerment for all residents (37). Future neighborhood developments can engage community residents and offer decision making power to create a vision for neighborhood change that is shared among all residents. The NYC Department of Health and Mental Hygiene, through the Neighborhood Health Action Centers, has mechanisms in place and physical space to implement such community interventions to neighborhoods where gentrification is currently taking place. With locations in East Harlem, Brownsville, and Tremont, and with many established community partners, the Neighborhood Health Action Centers, while not positioned to address the structural causes of inequalities described, can be useful tools in mitigating gentrification-associated health outcomes in the most impacted communities (65). Further NYC DOHMH gentrification research should include investigating health impacts of policies that directly address structural inequalities.”

• The majority of the data used in this manuscript can be found in the Open Science Framework Data Repository at https://osf.io/szv36. The repository contains a document titled: Gentrifcation_Manuscript_Datasets.doc that explains the contents of the files and original data sources for the Census and American Community Survey data used to classify neighborhoods into gentrification categories and Census Bureau population estimates. 

• The Community Health Survey data used to estimate the prevalence of serious psychological distress are available, albeit limited, at https://www1.nyc.gov/site/doh/data/data-sets/community-health-survey-public-use-data.page. The Community Health Survey Public Use Data is available at the city-wide level. In order to access the UHF level data as used in the manuscript, you would need to contact epidatarequest@health.nyc.gov to arrange a Data Use Agreement.

3. We note that Figure 1 in your submission contain [map/satellite] images which may be copyrighted. All PLOS content is published under the Creative Commons Attribution License (CC BY 4.0), which means that the manuscript, images, and Supporting Information files will be freely available online, and any third party is permitted to access, download, copy, distribute, and use these materials in any way, even commercially, with proper attribution. For these reasons, we cannot publish previously copyrighted maps or satellite images Neighborhood conditions influence people’s health; sustaining healthy neighborhoods is a priority of the New York City Health Department. Gentrification is characterized by rapid development in historically disinvested neighborhoods. The burden of gentrification, including increased living expenses, and disrupted social networks, disproportionally impacts certain residents. We examined psychological distress in gentrifying New York City neighborhoods to describe the association of gentrification and mental health and to target health promotion interventions. We categorized New York City neighborhoods during 2000–2017 as hypergentrifying, gentrifying, and not-gentrifying using a modified New York University Furman Center index. Neighborhoods with >100% rent growth were hypergentrifying; neighborhoods with greater than median and ≤100% rent growth were gentrifying; and neighborhoods with less than median rent growth were not-gentrifying. We calculated prevalence of nonspecific psychological distress among adults using 10 New York City Community Health Surveys during 2002–2015. Using joinpoint and survey-weighted logistic regression, we analyzed serious psychological distress prevalence trends during 2002–2015 by gentrification level, stratified by race/ethnicity. Among 42 neighborhoods, 7 were hypergentrifying, 7 were gentrifying, and 28 were not gentrifying. In hypergentrifying neighborhoods, serious psychological distress prevalence decreased among Whites (8.1% to 2.3%, β=-0.77, P=0.02) and was stable among Blacks (4.6% to 6.9%, β=-0.01, P=0.95) and Latinos (11.9% to 10.4%, β=-0.16, P=0.31). As neighborhoods gentrified, different populations were affected differently. Psychological distress decreased among white populations in hypergentrifying neighborhoods, no similar reductions were observed among black and Latino populations. This analysis highlights potential unequal mental health impacts that can be associated with gentrification-related neighborhood changes. Our findings will be used to target health promotion activities to strengthen community resilience and to ultimately guide urban development policiescreated using proprietary data, such as Google software (Google Maps, Street View, and Earth). For more information, see our copyright guidelines: http://journals.plos.org/plosone/s/licenses-and-copyright.

• We own Figure 1 and it is not copyrighted by another entity. This figure was created using a UHF shapefile that is produced by the NYC DOHMH and available for public use. It can be found here: https://www1.nyc.gov/site/doh/data/data-sets/maps-gis-data-files-for-download.page and in the Open Science Framework Data Repository at https://osf.io/szv36. The other data used to create this figure were the Census and ACS data described in the manuscript and in the repository. The figure was created using just these data sets and a combination of two opensource software: R and QGIS.

---

## [Decision Letter · Decision Letter 1]

8 Nov 2022

PONE-D-21-39108R1Can changing neighborhoods influence mental health? An ecological analysis of gentrification and neighborhood-level serious psychological distress — New York City, 2002–2015PLOS ONE

Dear Dr. Cavalier,

Thank you for submitting your manuscript to PLOS ONE. After careful consideration, we feel that it has merit but does not fully meet PLOS ONE’s publication criteria as it currently stands. Therefore, we invite you to submit a revised version of the manuscript that addresses the points raised during the review process. Whilst the most recent version of this manuscript features signifcant improvements, there are several important concerns raised by the reviewers that should be addressed prior to publication. It is strongly recommended that the authors revise the manuscript in accordance with these recommendations, including those posed by Reviewer 3, prior to rebusmission.

We look forward to receiving your revised manuscript.

Kind regards,

Blake Byron Walker, Ph.D.

Academic Editor

PLOS ONE

Reviewers' comments:

Reviewer's Responses to Questions

**Comments to the Author**

1. If the authors have adequately addressed your comments raised in a previous round of review and you feel that this manuscript is now acceptable for publication, you may indicate that here to bypass the “Comments to the Author” section, enter your conflict of interest statement in the “Confidential to Editor” section, and submit your "Accept" recommendation.

Reviewer #1: All comments have been addressed

Reviewer #2: (No Response)

Reviewer #3: (No Response)

2. Is the manuscript technically sound, and do the data support the conclusions?

Reviewer #1: Yes

Reviewer #2: Partly

Reviewer #3: Partly

3. Has the statistical analysis been performed appropriately and rigorously? 

Reviewer #1: Yes

Reviewer #2: Yes

Reviewer #3: Yes

4. Have the authors made all data underlying the findings in their manuscript fully available?

Reviewer #1: Yes

Reviewer #2: Yes

Reviewer #3: No

5. Is the manuscript presented in an intelligible fashion and written in standard English?

Reviewer #1: Yes

Reviewer #2: Yes

Reviewer #3: Yes

6. Review Comments to the Author

Reviewer #1: The authors have addressed all comments satisfactory. I only have one remaining concern.

Regarding the use of quotes from participants in an external publication, I still think these should be removed. While I understand the authors' intent of representing community perspectives which could not be captured by the quantitative data analyzed for the study, what should be presented here is the relevant findings from those external studies as interpreted by the authors of the external studies. To take a quotation from the results section of a qualitative study and publish it separately takes the citation away from the context in which it was analyzed. To use a piece of data like this also may violate research ethics in that the participant whose quote it is would have given permission to those authors to use their quote anonymously in their article. But they would not have given permission for the quote to then be reproduced elsewhere. Therefore, I would like to insist that these quotes be removed.

Reviewer #2: The authors did a great job engaging with the previous reviewer feedback. I was quite impressed. I have a few comments and suggestions that the authors should consider.

Many of the references about gentrification and community are older. The authors should review recent research by Joseph Gibbons and colleagues to these references.

The statement that gentrification measure is debated needs a citation. The authors should consider Barton (2016 Urban Studies) and Brown-Saracino (2017 Annual Review).

Related, given the variation in measurement, why was this particular strategy used? Were any other measures assessed to check for sensitivity? Barton and Cohen (Social Science Research) show that the measurement strategy has important implications. I don't think a second strategy needs to be reported, but the authors should engage with this issue in a slightly more thorough fashion.

The materials and methods section was underdeveloped. This could be expanded just a little to highlight/remind the reader of how this manuscript contributes to research on gentrification with health.

I understand the challenge of studying gentrification in New York City. In particular, having to use larger units of analysis such as UHFs or sub-boroughs. UHFs are large areas and gentrification typically occurs in a small part of a neighborhood. Barton (Crime and Delinquency) looked at the percent of tracts within subboroughs that gentrified. It's not a great measure, but it is more precise than simply looking at gentrification in the larger units. In short, there needs to be more discussion of the units of analysis to better justify these units and how gentrification is measured.

The first paragraph of the discussion should include a reminder of how this study contributed to previous research. This is a minor issue, but it helps with the flow of the manuscript.

Overall, I think this is an interesting study and has the potential to make contributions to research on gentrification and health. The authors did a great job engaging with the previous reviewer feedback and I hope they find these comments helpful.

Reviewer #3: This is a revised manuscript (MS) aiming to examine serious psychological distress as an indicator of gentrification’s impact on mental health in New York City (NYC) and whether the effect on mental health varies with race/ethnicity. To address this aim, the authors used data from two sources: The NYC Community Health Survey (CHS; years 2002, 2003, 2005, 2006, 2008, 2009, 2010, 2012, 2013, and 2015) and the American Community Survey (5-year estimates: 2013-2017).

Specific aims:

1) The abstract should provide the complete aim of the paper, i.e., variation by race/ethnicity.

2) The study is ecological as per the title. However, the language in the abstract goes back and forth from individuals to populations (lines 50-54).

3) The introduction does not provide a rationale for the use of psychological distress or why a difference by race/ethnicity should be expected.

4) The aims could be more clearly stated as they are lost in the last paragraph pf the introduction (lines 93-108).

5) The methods section has an unusual presentation and is hard to follow. For instance, the description of the datasets are combined with the descriptions of the exposure and outcome.

6) There is no justification for the years of the data included for the outcome.

7) The statistical analysis section could be more succinct and better organized to reflect the aims. In fact, as is, the statistical analysis description seems disconnected from the aims: there was never mentioning of trend over time or housing mobility in the aims.

8) Why wasn’t there a Table 1 describing details for the datasets used, i.e., number of neighborhood gentrified, mean (median) number of people per neighborhoods, prevalence of psychological distress by neighborhoods, demographics of the neighborhoods based on gentrification, etc.?

9) Where is the data presented in lines 248-256? If not in tables, please say so.

10) Figure 3 needs legends for the axes.

11) The Discussion section is too long and a bit out of scope of the aims.

12) The limitations should include the effect of MAUP, uncertainty geographic context problem and spatial polygamy.

7. PLOS authors have the option to publish the peer review history of their article (what does this mean?). If published, this will include your full peer review and any attached files.

Reviewer #1: No

Reviewer #2: No

Reviewer #3: No

---

## [Author Response · Author response to Decision Letter 1]

22 Dec 2022

Reviewer #1: The authors have addressed all comments satisfactory. I only have one remaining concern.

Regarding the use of quotes from participants in an external publication, I still think these should be removed. While I understand the authors' intent of representing community perspectives which could not be captured by the quantitative data analyzed for the study, what should be presented here is the relevant findings from those external studies as interpreted by the authors of the external studies. To take a quotation from the results section of a qualitative study and publish it separately takes the citation away from the context in which it was analyzed. To use a piece of data like this also may violate research ethics in that the participant whose quote it is would have given permission to those authors to use their quote anonymously in their article. But they would not have given permission for the quote to then be reproduced elsewhere. Therefore, I would like to insist that these quotes be removed.

Thank you for this feedback, the reviewer makes a good point. The two direct quotes from qualitative research studies were removed from the manuscript. 

Reviewer #2: The authors did a great job engaging with the previous reviewer feedback. I was quite impressed. I have a few comments and suggestions that the authors should consider.

Thank you very much for your review and support, we truly think this helped to improve the manuscript. 

Many of the references about gentrification and community are older. The authors should review recent research by Joseph Gibbons and colleagues to these references.

Thank you for this recommendation, we have included updated references by author Joseph Gibbons and colleagues, including the following two references:

• Gibbons J. Are gentrifying neighborhoods more stressful? A multilevel analysis of self-rated stress. SSM-population health. 2019 Apr 1;7:100358. 

• Gibbons J, Barton MS, Reling TT. Do gentrifying neighbourhoods have less community? Evidence from Philadelphia. Urban Studies. 2020 May;57(6):1143-63.

The statement that gentrification measure is debated needs a citation. The authors should consider Barton (2016 Urban Studies) and Brown-Saracino (2017 Annual Review).

Thank you for these recommended citations, these two references were added accordingly.

Related, given the variation in measurement, why was this particular strategy used? Were any other measures assessed to check for sensitivity? Barton and Cohen (Social Science Research) show that the measurement strategy has important implications. I don't think a second strategy needs to be reported, but the authors should engage with this issue in a slightly more thorough fashion.

On line 98, we have included additional rationale for the selection of the gentrification measure used in this study “We selected the Furman Center definition to build on this Center’s substantial foundation of scholarship on gentrification in NYC, and because the definition relied on multiple economic factors. Gentrification definitions that incorporate social criteria, such as educational attainment and single-parent households, might unfairly put the onus of neighborhood change on the neighborhood’s community members.”

The materials and methods section was underdeveloped. This could be expanded just a little to highlight/remind the reader of how this manuscript contributes to research on gentrification with health.

Thank you for this suggestion. Initially in the abstract on line 57 we had written “This analysis highlights potential unequal mental health impacts that can be associated with gentrification-related neighborhood changes.“ However, we completely agreed that we needed to more clearly highlight how this manuscript contributes to research on gentrification and health. 

To rectify this, in the introduction we added a section to more carefully describe our aims in order to help clarify and contextualize our methods section. On line 123 we wrote “In this study we aimed to characterize trends in gentrification and neighborhood-level serious psychological distress over time, and how the relationship varied by racial/ethnic populations. Given the relatedness between racism and gentrification reported in previous literature (28), we hypothesized that the population composition of original residents versus new residents, and thus who are exposed to gentrification, varied by race/ethnicity. We explored the validity of this assumption in an ancillary analysis using housing mobility data”

Lastly, we included additional language in the Discussion section which highlights how our findings contribute to the literature. We have added at line 360 the following text “These findings enhance our understanding of gentrification and health by showing how neighborhood population mental health in NYC has changed over time as neighborhoods themselves have changed, and by reinforcing the notion that gentrification has differential racial/ethnic impacts and can exacerbate health disparities.”

I understand the challenge of studying gentrification in New York City. In particular, having to use larger units of analysis such as UHFs or sub-boroughs. UHFs are large areas and gentrification typically occurs in a small part of a neighborhood. Barton (Crime and Delinquency) looked at the percent of tracts within subboroughs that gentrified. It's not a great measure, but it is more precise than simply looking at gentrification in the larger units. In short, there needs to be more discussion of the units of analysis to better justify these units and how gentrification is measured.

Thank you for pointing this out, it is a great point. While the gentrification method described in the Barton paper is smart and may provide more granularity within the bigger geography, the NYC DOHMH often uses UHF for community and neighborhood health metrics because this geography has meaningful neighborhood boundaries. The text has been updated in the discussion to reflect this. Lines 492-505 now state “Additionally, the gentrification exposure might not be acting uniformly within neighborhoods, and some neighborhoods might be influenced by gentrification at different times, speeds, and spatial scales. The causally relevant geographic unit important for mental health outcomes is unknown, and mental health may also be influenced by the neighborhoods where people work, attend school, or otherwise spend time aside from their residential neighborhood. Thus, the study is susceptible to the uncertain geographic context problem and spatial polygamy, respectively (65). We examined gentrification at the UHF level, as this is the smallest geography for which our survey data can provide representative prevalence estimates, however, since gentrification can occur on a hyper-local spatial scale, gentrification might be heterogeneous within a given UHF (Fig 1). Nonetheless, UHFs, and the Community Districts they were designed to approximate, represent meaningful neighborhood boundaries. Residents in these neighborhoods share community board members who advise important decisions on resource allocation, land use, zoning, and other city planning and development projects, all of which are relevant to gentrification (29, 66).”

The first paragraph of the discussion should include a reminder of how this study contributed to previous research. This is a minor issue, but it helps with the flow of the manuscript.

Thank you for this suggestion, we have added a sentence at the end of the first paragraph of the discussion that clarifies how our work contributes to prior research. On line 360 we say “These findings enhance our understanding of gentrification and health by showing how neighborhood population mental health in NYC has changed over time as neighborhoods themselves have changed, and by reinforcing the notion that gentrification has differential racial/ethnic impacts and can exacerbate health disparities.”

Overall, I think this is an interesting study and has the potential to make contributions to research on gentrification and health. The authors did a great job engaging with the previous reviewer feedback and I hope they find these comments helpful.

Thank you so much! 

Reviewer #3: This is a revised manuscript (MS) aiming to examine serious psychological distress as an indicator of gentrification’s impact on mental health in New York City (NYC) and whether the effect on mental health varies with race/ethnicity. To address this aim, the authors used data from two sources: The NYC Community Health Survey (CHS; years 2002, 2003, 2005, 2006, 2008, 2009, 2010, 2012, 2013, and 2015) and the American Community Survey (5-year estimates: 2013-2017).

Specific aims:

1) The abstract should provide the complete aim of the paper, i.e., variation by race/ethnicity.

Thank you for this recommendation. In the abstract on lines 39-42 we revised our stated aim by saying “To ultimately target health promotion interventions, we examined serious psychological distress time trends in gentrifying NYC neighborhoods to describe the association of gentrification and mental health overall and stratified by race and ethnicity.”

2) The study is ecological as per the title. However, the language in the abstract goes back and forth from individuals to populations (lines 50-54).

Thank you for this recommendation. Here in the abstract, and throughout the manuscript language was revised to ensure that analyses reflected neighborhood-level population estimates and not estimates of individuals.

3) The introduction does not provide a rationale for the use of psychological distress or why a difference by race/ethnicity should be expected.

Thank you for these comments.

Psychological distress: Lines 104-105 state “We chose to examine serious psychological distress as an indicator of gentrification’s impact on mental health based on literature evidence and mechanistic plausibility (12, 21-24).” We have added at lines 105-107 “We chose to use serious psychological distress as it captures a variety of mental health disorders in a single metric at a population level (25).

Difference by race/ethnicity: Throughout the introduction we describe how gentrification may intersect with race and ethnicity and how the practices creating gentrified neighborhoods are deeply embedded in racism. We have added (lines 115-117) to emphasize how this informed our methods where we state “We examined serious psychological distress stratified by race/ethnicity, as we expected mental health outcomes related to gentrification to vary by race”. Which is followed by examples in lines 117-121 about exclusion and displacement related to race. We have additionally added at line 124 “Given the relatedness between racism and gentrification reported in previous literature (28), we hypothesized that the population composition of original residents versus new residents, and thus who are exposed to gentrification, varied by race/ethnicity.”

4) The aims could be more clearly stated as they are lost in the last paragraph of the introduction (lines 93-108).

Thank you for this recommendation, for increased clarity we added the following text at lines 123-128 “In this study we aimed to characterize trends in gentrification and neighborhood-level serious psychological distress over time, and how the relationship varied by racial/ethnic populations. Given the relatedness between racism and gentrification reported in previous literature (28), we hypothesized that the population composition of original residents versus new residents, and thus who are exposed to gentrification, varied by race/ethnicity. We explored the validity of this assumption in an ancillary analysis using housing mobility data.”

5) The methods section has an unusual presentation and is hard to follow. For instance, the description of the datasets are combined with the descriptions of the exposure and outcome.

While we understand where the reviewer is coming from, we chose to describe the data sets with the exposure and outcome variable definitions, as separate data sets were used for each metric. Typically, if the exposure and outcome were coming from the same data set, we would describe the data set first and then dive into exposure and outcome variables separately. In this specific situation, this presentation would be confusing and require us to repeat ourselves in sections about data sets and variables. Given the limited character space, we respectfully choose to keep the methods how they currently are presented.

6) There is no justification for the years of the data included for the outcome.

We added a sentence to line 176 in the methods section that clarified that “The timeframe examined in this study was, in part, shaped by the available outcome data on psychological stress.” 

In addition, we added a sentence in the abstract on line 45 that clarifies that “To temporally align neighborhood categorization closely with neighborhood-level measurement of serious psychological distress, data during 2000–2017 were used to classify neighborhood type.” 

7) The statistical analysis section could be more succinct and better organized to reflect the aims. In fact, as is, the statistical analysis description seems disconnected from the aims: there was never mentioning of trend over time or housing mobility in the aims.

Thank you for pointing this out. We reordered the sentences in the section on statistical analysis so that it first describes the analysis on time trends, linking to our primary aim. On line 248 the text reads “For time trend analyses, statistical significance of trends was determined if regression model estimate P-values were <0.05.” 

After this, in the second sentence of this statistical analysis paragraph, we describe the statistical analysis related to our secondary aim which examined housing mobility. On line 249 we wrote “Pairwise T-tests were used to compare different strata using a reference category for the 2015 Community Health Survey data on serious psychological distress, and 2017 Community Health Survey data on housing mobility.” 

Also in the methods, words/phrases were reworked or deleted to be more succinct. For example, in the section describing time trend analyses we removed “also known as change point or segmented regression”.

We hope that these changes, in conjunction with the more clearly articulated aims in the introduction in lines 123-128 will help readers more easily understand our methodology. Now our aims are written as follows “In this study we aimed to characterize trends in gentrification and neighborhood-level serious psychological distress over time, and how the relationship varied by racial/ethnic populations. Given the relatedness between racism and gentrification, we reasonably assumed that the population composition of original residents versus new residents, and thus who are exposed to gentrification, varied by race/ethnicity. We explored the validity of this assumption in an ancillary analysis using housing mobility data.”

8) Why wasn’t there a Table 1 describing details for the datasets used, i.e., number of neighborhood gentrified, mean (median) number of people per neighborhoods, prevalence of psychological distress by neighborhoods, demographics of the neighborhoods based on gentrification, etc.?

Thank you for this recommendation. The geography we used in our analysis was the United Hospital Fund (UHF), which is described in our manuscript on line 152 “as 42 adjoining zip code areas intended to approximate NYC community planning districts, with roughly 200,000 individuals residing within each UHF.“ We added an additional sentence on line 153 directing readers to the community profile websites in case they sought out additional neighborhood information “Detailed neighborhood profiles are described online by the NYC Health Department.” These neighborhood profiles have been created by the NYC Department of Health and Mental Hygiene and they describe in detail the characteristics and health metrics for each NYC neighborhood. These can be found here: https://www.nyc.gov/site/doh/data/data-publications/profiles.page

We originally summarized the neighborhood characterization in writing in the results section and graphically in Figure 1. The geographic illustration offered meaningful context both for people familiar with and those unfamiliar with NYC neighborhoods. Stemming from this recommendation, however, we have created an S1 Table as supplementary material for those readers who want to see additional detail on how each neighborhood met the different categorization criteria. 

9) Where is the data presented in lines 248-256? If not in tables, please say so.

We added a new table describing the criteria for neighborhood classification and it was added to the supplementary materials. On line 264 we revised the text to read “A summary of the gentrification criteria by neighborhood can be found in the supplementary material in S1 Table.“

10) Figure 3 needs legends for the axes.

Given that there can be subtle differences in the seven shades of gray used in this figure, we felt that directly labeling the bar graphs would be easier for readers to interpret than creating a legend. 

11) The Discussion section is too long and a bit out of scope of the aims.

Thank you for this feedback. We have addressed this recommendation in multiple ways. 

• We removed quotes from participants of qualitative research, as these were not referencing any of the aims we stated. 

• We updated the aims in the introduction section to more explicitly include the housing mobility analysis. This brings greater relevance of the issue of housing mobility described in the 4th paragraph of the discussion. 

• Other sentences were removed that we thought were less relevant to specific aims. These include the sentence reading “Older adults living in neighborhoods undergoing gentrification have more symptoms of depression and anxiety compared with counterparts in high-income neighborhoods”

12) The limitations should include the effect of MAUP, uncertainty geographic context problem and spatial polygamy.

Thank you for pointing this out. Lines 489-498 now include a discussion of these concepts. 

“An ecological study design inherently limits inferences to an aggregated neighborhood-level, and we cannot infer causality of gentrification to individual-level health outcomes in this study. Related, the modifiable areal unit problem, also results from the aggregation of survey data to UHF geography, as the aggregate values are influenced by what geography is chosen (64). Additionally, the gentrification exposure might not be acting uniformly within neighborhoods, and some neighborhoods might be influenced by gentrification at different times, speeds, and spatial scales. The causally relevant geographic unit important for mental health outcomes is unknown, and mental health may also be influenced by the neighborhoods where people work, attend school, or otherwise spend time aside from their residential neighborhood. Thus, the study is susceptible to the uncertain geographic context problem and spatial polygamy, respectively (65).”

---

## [Editor Report · Decision Letter 2]

11 Jan 2023

PONE-D-21-39108R2Can changing neighborhoods influence mental health? An ecological analysis of gentrification and neighborhood-level serious psychological distress — New York City, 2002–2015PLOS ONE

Dear Dr. Cavalier,

Thank you for submitting your manuscript to PLOS ONE. After careful consideration, we feel that it has merit but does not fully meet PLOS ONE’s publication criteria as it currently stands. Therefore, we invite you to submit a revised version of the manuscript that addresses the points raised during the review process. There are several remaining concerns raised by reviewers that should be addressed in full. Concerning the removal of quotations, this depends entirely upon both (a) whether study participants agreed to be directly quoted in materials resulting from the study and (b) whether the appropriate ethics approvals including a detailed description of the qualitative data collection and data handling were secured, if applicable at your institute. If both of these criteria are fulfilled, please do retain the quotes, as they provide interesting contextual data; however, both points (a) and (b) should be explicitly addressed in the revised manuscript text.

We look forward to receiving your revised manuscript.

Kind regards,

Blake Byron Walker, Ph.D.

Academic Editor

PLOS ONE
---

## [Author Response · Author response to Decision Letter 2]

22 Feb 2023

There are several remaining concerns raised by reviewers that should be addressed in full. Concerning the removal of quotations, this depends entirely upon both (a) whether study participants agreed to be directly quoted in materials resulting from the study and (b) whether the appropriate ethics approvals including a detailed description of the qualitative data collection and data handling were secured, if applicable at your institute. If both of these criteria are fulfilled, please do retain the quotes, as they provide interesting contextual data; however, both points (a) and (b) should be explicitly addressed in the revised manuscript text.

We believe that the reviewer who suggested the removal of the quotes made a compelling argument that it is not standard practice to include participant quotes from other studies, and that this practice removes the quote from the rich context the qualitative study provides. Thus, we do not feel inclined to attempt to contact the original authors and request the additional ethics approvals needed to keep the quotes in our paper. 

Journal Requirements:

Thank you for your close attention to our reference list. We are grateful for this feedback, because these corrections help to attribute information more accurately to current gentrification and health literature. We have made the following changes to ensure appropriateness and accuracy of citations.

• In the discussion on line 374, reference 43 was corrected to reference 42 (Brummet et al) – this paper indicates that some original residents might benefit from neighborhood gentrification. No downstream numbering was impacted due to this change, because reference 43 (Izenberg et al) was cited again in the same sentence – a paper that describes that gentrification can harm original residents.

• The Anguelovski reference (originally reference 57) was removed because people in Barcelona are not considered Latinos (in general Spanish speaking from Spain are considered Hispanic, but we had another reference already for this sentence – reference 56- and we thought it would be more confusing to keep reference 57 and/or highlight this difference)

• The Glaeser reference (originally reference 61) was removed. This online report posted by the National Bureau of Economic Research was very recently revised in February 2023. The report no longer includes the example about the Boyle Heights LA neighborhood, that we had originally included in our discussion. To make our point that gentrification solely based on economics does not account for certain psychosocial changes we site other references, including a review article by Bhavsar et al (now reference 59), as well as Lance Freeman’s book ‘There goes the hood: Views on gentrification from the ground up’ (now reference 60). 

Reviewers' comments:

Done. PACE made a few adjustments to the figures including flattening, compressing, adjusting dimensions, and removing the Alpha channel. The adjusted figures were downloaded from PACE and will be the files submitted with this revision.

---

## [Editor Report · Decision Letter 3]

6 Mar 2023

Can changing neighborhoods influence mental health? An ecological analysis of gentrification and neighborhood-level serious psychological distress — New York City, 2002–2015

PONE-D-21-39108R3

Dear Dr. Cavalier,

We’re pleased to inform you that your manuscript has been judged scientifically suitable for publication and will be formally accepted for publication once it meets all outstanding technical requirements.

Kind regards,

Blake Byron Walker, Ph.D.

Academic Editor

PLOS ONE

---

## [Editor Report · Acceptance letter]

9 Mar 2023

PONE-D-21-39108R3 

Can changing neighborhoods influence mental health? An ecological analysis of gentrification and neighborhood-level serious psychological distress — New York City, 2002–2015 

Dear Dr. Cavalier:

I'm pleased to inform you that your manuscript has been deemed suitable for publication in PLOS ONE. Congratulations! Your manuscript is now with our production department. 

Kind regards, 

on behalf of

Prof. Dr. Blake Byron Walker 

Academic Editor

PLOS ONE